# The Course and Surgical Treatment of Acute Appendicitis during the First and Second Wave of the COVID-19 Pandemic: A Retrospective Analysis in University Affiliated Hospital in Latvia

**DOI:** 10.3390/medicina59020295

**Published:** 2023-02-05

**Authors:** Anna Marija Lescinska, Elza Sondore, Margarita Ptasnuka, Maksims Mukans, Haralds Plaudis

**Affiliations:** 1Faculty of Medicine, University of Latvia, LV-1004 Riga, Latvia; 2Faculty of Medicine, Riga Stradins University, LV-1007 Riga, Latvia; 3Department of General and Emergency Surgery, Riga East Clinical University Hospital, LV-1038 Riga, Latvia

**Keywords:** COVID-19, acute appendicitis, surgery

## Abstract

*Background and Objectives:* Acute appendicitis is the most common abdominal emergency requiring surgery and it has an estimated lifetime risk of 6.7 to 8.6%. The COVID-19 pandemic has transformed medical care worldwide, influencing diagnostic tactics, treatment modalities and outcomes. Our study aims to compare and analyze management of acute appendicitis before and during the first and second waves of the pandemic. *Materials and Methods:* Patients suffering acute appendicitis were enrolled retrospectively in a single-center study for a 10-month period before the pandemic (pre-COVID-19 period: 1 March to 31 December 2019) and during the pandemic (COVID-19 period: 1 March to 31 December 2020). The total number of patients, disease severity, diagnostic methods, complications, length of hospitalization and outcomes were analyzed. *Results:* A total number of 863 patients were included, 454 patients in the pre-COVID-19 period and 409 patients in the COVID-19 period. Compared to the pre-COVID-19 period, the number of complicated appendicitis increased in the COVID-19 period (24.4% to 37.2%; *p* < 0.001). The proportion of laparoscopic appendectomies increased during the COVID-19 period but did not show statistically significant differences between periods. In both time periods, we found that open technique was the chosen surgical approach more frequently in elderly patients (*p* < 0.001). Generalized peritonitis was significantly more common during the COVID-19 period (3.5% vs. 6.1%, *p* < 0.001). The postoperative course of patients was similar in the pre-COVID-19 period and during the COVID-19 period, with no significant differences in ICU admissions, overall hospital stay or morbidity. *Conclusions:* The COVID-19 pandemic has led to a significant increase in complicated forms of acute appendicitis; however, no significant impact was observed in terms of diagnostic or treatment approach.

## 1. Introduction

On 31 December 2019, the Chinese health authorities reported an outbreak of pneumonia of unknown etiology in Wuhan City, China [1,2] and on 9 January 2020 the Chinese Center for Disease Control identified it as a new coronavirus SARS-CoV-2 [3,4]. On 11 February 2020, the World Health Organization (WHO) gave the name “COVID-19” (Corona Virus Disease) to the associated respiratory disease and, on 11 March 2020, the WHO declared the COVID-19 outbreak a pandemic [5].

Latvia is a country in the Baltic region of Eastern Europe with a population of 1.85 million. The first COVID-19 patient in Latvia was diagnosed on 2 March and 197 persons were found to be infected by 24 March 2020 [6]. The state of emergency was declared on 13 March to restrict the spread of COVID-19. In Latvia, at the first wave of the pandemic, the overall infection rate was registered lower than in most European countries, but it rapidly increased reaching a 14-day cumulative rate of 583.6 newly reported COVID-19 cases per 100,000 inhabitants at the second wave of infection [7]. Like in many other countries, to prevent the breakdown of the national health system, the Latvian Government and the State Operative Medical Commission developed a strategy to prioritize emergency conditions and postponed many scheduled health care services, as well as all non-essential face-to-face outpatient appointments across all medical specialties.

Unlike in emergencies with known dynamics, the sudden outbreak of the pandemic, the lack of specific knowledge of the COVID-19 infection and evidence-based medicine guidelines made it extremely difficult to plan the first phase steps of the pandemic. At the onset of the COVID-19 pandemic, the planning of surgical activities faced several challenges. It included making the necessary resources such as ventilators and intensive care beds primarily available for treating COVID-19 positive patients; restructuring departments and shifts of medical staff; and rescheduling all elective surgical activities.

Acute appendicitis is a common cause of acute abdomen and is considered to be one of the most frequent surgical emergencies worldwide, with an estimated lifetime risk of 6.7 to 8.6% [8]. Most patients can achieve good therapeutic results by timely appendectomy. Patients with perforated appendicitis, however, have a mortality rate of up to 5% [9,10]. While the current treatment recommendation for uncomplicated acute appendicitis is surgical appendectomy, the non-surgical alternative of medical management by antibiotic therapy has been widely discussed in the literature in context of the COVID-19 pandemic. So far, the main limitation of conservative treatment of uncomplicated acute appendicitis is the risk of appendicitis recurrence. This treatment option allows to reduce the intra-hospital stay and hospital overload in the situation of a health crisis [11]. In Latvia, operative approach is recommended for acute appendicitis.

The COVID-19 pandemic has transformed medical care worldwide, influencing diagnostic tactics, treatment modalities and outcomes, and has even caused public health crises [11]. In the light of the global pandemic of COVID-19, the clinical presentation, complication rate and management of acute appendicitis were influenced significantly—because of the epidemiological situation and the occupation of hospital beds by COVID-19 patients [12]. Consequently, the aim of our study was to compare the characteristics, diagnostics, management, and outcomes of patients with acute appendicitis in Riga East Clinical University Hospital during the COVID-19 pandemic in 2020 compared to a pre-COVID-19 period.

## 2. Materials and Methods

This retrospective cohort analysis is undertaken at Riga East University Hospital in Latvia that provides tertiary center healthcare to the entire population of Riga and its surrounding area, which is approximately 700,000 [13]. During the year 2020, there were 72,723 Emergency Department visits, and 41,781 surgical procedures were performed. Since the initial outbreak of COVID-19 pandemic, this hospital was designated to serve as the main “COVID hospital” for COVID-19 positive patients in Latvia.

### 2.1. Emergency Department Pathway for Patients with Acute Appendicitis

The diagnosis of acute appendicitis in the hospital is based on clinical assessment coupled with laboratory data and pre-operative imaging study, such as ultrasound (US) and/or computed tomography (CT). From the local emergency protocols of the hospital, US is a first-line diagnostic modality in the case of suspected acute appendicitis, followed by CT scanning in cases that remain negative or inconclusive after US examination. Our hospital follows the surgical treatment strategy, while conservative management of acute appendicitis is chosen only in exceptional cases. Of note, the type of surgical approach is chosen by the surgeon, with laparoscopy often the method of choice. The operating team includes the chief resident accompanied by a resident in an earlier training stage. If the surgery becomes technically challenging, a board-certified surgeon is asked for a consultation or to undertake the surgery. To prevent intrahospital COVID-19 transmission, all patients in 2020 were operated only after receiving the COVID-19 virus PCR test results as required for all emergency admissions. In our hospital, patients were tested with both Rapid and long PCR COVID-19 tests, depending on the urgency of clinical condition.

All adult patients (≥18 years old) who were admitted to the Department of General and Emergency Surgery of Riga East Clinical University Hospital and underwent appendectomy for acute appendicitis between March and December 2020 (COVID-19 period) and in the same consecutive months in 2019 (pre-COVID-19 period) were included in the study, using International Classification of Diseases (ICD-10) diagnosis codes for acute appendicitis. Based upon the available definition, uncomplicated appendicitis was defined as an inflamed appendix without signs of necrosis or perforation, and without local or systemic complications. Complicated appendicitis was defined as an inflammation of appendix with gangrene or perforation, abscess, local or generalized peritonitis causing systemic response [14].

Variables collected included demographic data (age, gender), preoperative imaging (US, CT), severity of acute appendicitis, surgical approach, time between arrival at the Emergency Department and the start of the operation. Furthermore, postoperative parameters such as overall hospital stay, ICU admission and mortality were analyzed. All collected data were compared between the COVID-19 period and the pre-COVID-19 period.

### 2.2. Statistical Analysis

Data statistical analysis was performed using the SPSS program (IBM SPSS Statistics. v. 23.0. Chicago. IL. USA). The normal distribution of interval data was tested using the Shapiro—Wilk test. As normal distribution was not found, only non-parametric data processing methods were applied in further analysis. Quantitative data were presented using median (Me) and interquartile range (IQR). According to the results, the Mann–Whitney U test was used to compare the interval data, the Wilcoxon test was used to compare the dynamics of the interval data. The Spearman rho test was used for the correlation of interval or ordinal data. Interpretation of correlation coefficients: up to |±0.30| weak. |±0.31| medium strong and strong above |±0.65|. Pearson chi square and Fisher’s exact tests were used for the rank data proportions comparison. Adjusted residuals (AR) were used with Pearson Chi square tests in cases with multiple subgroups comparison. AR > +/−1.96 were considered statistically significant, *p* values < 0.05 were considered statistically significant. All tables and graphs were created using Microsoft Office Excel (365) program.

This paper has been drafted according to the Strengthening the Reporting of Observational studies in Epidemiology (STROBE) checklist [15].

## 3. Results

### 3.1. Patient Demographics

Overall, 863 patients were included in the study, 454 patients in the pre-COVID-19 period, and 409 patients in the COVID-19 period. During the COVID-19 period, all patients were tested for SARS-CoV-2 on admission, and none of the patients were positive for the disease. Patient demographics during COVID-19 period and the previous year are presented in Table 1. The median age of patients was constant and did not change between periods (35 y, IQR 51–26 vs. 35 y, IQR 49–27, *p* = 0.768). Additionally, there were no significant differences found regarding the patients’ gender or distribution of patients between age groups.

### 3.2. Diagnostic Procedures

Pre-operative imaging evaluation was performed for all patients with suspected appendicitis. US was the method of choice for 96.1% (829/863) of patients, while CT scan alone or as part of the examination plan after US was used in 30.6% (264/863). When each of these methods of imaging were compared between periods, no statistically significant difference was observed, see Table 2.

Comparing the number of CT scans performed in both years, no notable increase was observed.

### 3.3. Surgical Approach and Time until Treatment

Of all appendectomies, 79.3% (686/865) were performed laparoscopically, and 20.7% (179/865) were open or converted to open. The proportion of laparoscopic appendectomies increased during the COVID-19 period, but did not show statistically significant differences. Median time since admission until surgery initiation in the pre-COVID-19 period was 6.7 h (IQR 10.6–4.1), and 6.4 h (IQR 11.5–3.9) during the COVID-19 period, *p* = 0.661.

When comparing both time periods, open technique was the chosen surgical approach more frequently in elderly patients—over the age of 51 years, see Figure 1.

No statistical differences in time until surgery was noted between patients depending on radiological diagnostic modalities used, see Table 3.

### 3.4. Severity of Acute Appendicitis

During the COVID-19 period, a statistically higher proportion of patients presented with complicated appendicitis compared to the pre-COVID-19 period cohort. Consequently, the proportion of uncomplicated appendicitis dropped from 75.7% (345/456) in 2019 to 62.8% (257/409) in 2020, *p* < 0.001, see Table 1.

Analyzing only patients with complicated forms of acute appendicitis, local peritonitis was found during the surgery in 20.8% (n = 95) in the pre-COVID-19 period versus 31.1% (n = 127) during the COVID-19 period, *p* < 0.001. Furthermore, generalized peritonitis was significantly more common during the COVID-19 period (3.5% vs. 6.1%, *p* < 0.001).

The ratio of uncomplicated versus complicated appendicitis differed among the age groups. However, uncomplicated appendicitis was more common among younger patients, i.e., pre-COVID-19 period 18–24 y/o patients; COVID-19 period 18–34 y/o patients, see Figure 2.

### 3.5. Outcomes

The postoperative course of patients was similar in the pre-COVID-19 period and during the COVID-19 period, with no significant differences in ICU admissions, overall hospital stay or mortality, see Table 4. No differences were found in hospital stay between age groups in both periods. At the same time, our study showed a weak correlation between patients’ age and hospital stay; younger patients showed a shorter hospital stay compared to older patients (r = 0.312 vs. r = 0.348, *p* < 0.001). 

There was also a weak correlation between patients’ age, the severity of diagnosis and the length of hospitalization. During the pre-COVID-19 period, older patients spent more time in hospital with local peritonitis (pre-COVID-19 period: r = 0.405 *p* < 0.001 vs. COVID-19 period: r = 0.396 *p* = 0.001), however, during the COVID-19 period, with generalized peritonitis (pre-COVID-19 period: r = 0.48 *p* = 0.06 vs. COVID-19 period: r = 0.497; *p* = 0.012), see Table 5. One patient died due to pulmonary embolism during the pre-COVID-19 period, and there were no lethal outcomes during the COVID-19 period.

## 4. Discussion

The rapid spread of the COVID-19 pandemic led to crucial changes affecting all national healthcare systems worldwide. Quick implementation of lockdown, strict triage of patients and disease, reduction in elective surgeries or even a complete shift to surgical emergencies only caused new challenges for healthcare professionals globally.

An Italian multicenter study that analyzed data of surgical emergencies during COVID-19 outbreak showed a significant decrease in emergency surgical admissions and surgical operations with a fall of 45% and 41%, respectively [16]. This could be explained by the fact that people were forced to stay at home and visit the emergency department only in life-threatening situations. In addition, a few studies from Germany showed a decrease in the overall number of patients who underwent appendectomies after the onset of COVID-19 pandemic compared with the period before. The authors explain it by the reduction in the number of negative appendectomies and uncomplicated cases of acute appendicitis [17,18,19]. Lockdowns of the medical system were introduced in Latvia during the first and second waves of pandemics, epidemiological situation in the country remained stable, and the hospital was not overcrowded by COVID-19 patients. All resources such as the operating room, surgical department, radiology, and ICU were available, and patients were able to receive adequate care. Nevertheless, since the hospital beds were freed up for COVID-19 patients and a part of the emergencies transferred to other hospitals, the overall number of patients with acute appendicitis did not change significantly.

The diagnostic of acute appendicitis is predominantly based on a typical presentation of clinical symptoms and a combination of different radiological imaging (such as US and CT scan) modalities. There are guidelines that recommend certain diagnostic algorithms; however, adaptation and implementation of guidelines depend on available technological and financial resources [20]. In our hospital, abdominal US is the method of choice for primary assessment of patients with suspected appendicitis and CT scan is used selectively and mainly for differential diagnosis. It is known that implementation of strict COVID-19 testing algorithms can affect urgent availability of diagnostic procedures and waiting times for urgent surgeries [21]. However, our results showed that good collaboration with laboratory services and usage of rapid COVID-19 PCR tests did not affect diagnostic protocols and waiting times from admission to surgery.

Reported studies observed a 21% increase in the cases of perforated appendicitis [22]. Moreover, a 29% increase in the number of gangrenous appendicitis was noted and a direct correlation between the application of COVID-19 related restrictions and the severity of acute appendicitis is recognized. It reflects the fact that during the pandemic, patients requiring urgent surgical intervention were not seeking timely and appropriate care [22]. The same tendencies are observed in our study leading to an increase in complicated forms of acute appendicitis during the first and second waves of pandemic. We can speculate that this can be explained by patients’ fear of contracting COVID-19 in the hospital and misinformation about the availability of emergency care.

Surgical patients have unique risks due to COVID-19 infection—operating on patients with either asymptomatic or symptomatic COVID-19 increases the risk for perioperative morbidity and mortality. Another major challenge for surgery has been the need to stop non-urgent surgeries effectively and safely. Many patients with surgical diseases have not been operated on because of concerns of acquiring COVID-19 at hospitals. Non-urgent and non-emergency care has been delayed and has created a large backlog of patients who require surgical care [11].

The United Kingdom Intercollegiate guidance was published in late March and outlined recommendations for provision of surgery. It concluded that laparoscopic surgery, given its potential for aerosol generation, should be considered only in selected individuals where clinical benefit exceeds the potential risk of viral transmission. Analyzing current literature regarding the COVID-19 pandemic, non-operative management of certain conditions is favored and an open approach for acute appendicitis suggested as a suitable alternative when hospital resources are limited [22]. We did not observe an uncontrolled flow of COVID-19 patients to the emergency department, which gave us the possibility of not changing our diagnostic and treatment protocols. Surgery remained the method of choice for acute complicated and uncomplicated appendicitis. Of note, the number of complicated forms of the disease increased. So did the number of minimally invasive appendectomies, except in elderly patients.

Given the exceptional nature of the situation, there is a lack of evidence regarding the optimal management of patients with urgent surgical pathology. The information is changing, as the epidemiological knowledge of the disease advances. The establishment of multidisciplinary surgical committees that develop and implement action protocols is recommended.

### Strengths and Limitations

Although numerous studies during the COVID-19 pandemic period on acute appendicitis have been conducted, it remains difficult to compare them due to different quality and completeness of data. The main strength of this study is its large number of patients with complete clinical data included. Additionally, the inclusion period started at the national lockdown period in Latvia before any guidelines were adapted, providing a unique result of the pandemic impact on acute appendicitis.

This paper has some limitations that should be considered. First, we present retrospective data from a leading University affiliated hospital, where bias may exist in the decision for initial diagnostic and treatment management compared to other hospitals in Latvia. Second, we used ICD codes to identify patients with the diagnosis of acute appendicitis and it is possible that some patients have not been captured in our study. Moreover, ICD coding system is not applicable to analyze complete disease severity distribution according to grading scale. Additionally, we did not treat nor perform surgery on any COVID-19 positive patients with acute appendicitis during the early pandemic period.

## 5. Conclusions

This study describes the impact of the COVID-19 pandemic on treatment tactics of acute appendicitis in one of the largest Emergency Department in the Baltic states located in eastern Europe. Marked changes in healthcare system due to the COVID-19 pandemic and patient fear of contracting virus led to a significant increase in complicated forms of acute appendicitis. Due to strict in-hospital organizational procedures no significant impact was observed in terms of diagnostic protocols, surgical approach, and outcomes.

## Figures and Tables

**Figure 1 medicina-59-00295-f001:**
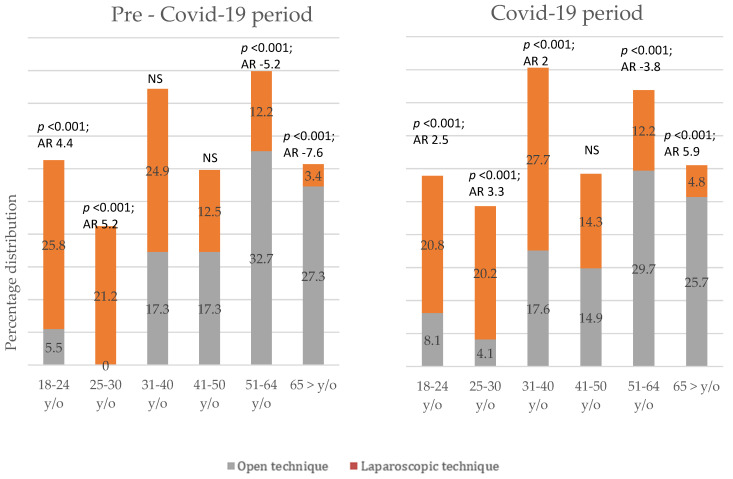
Distribution of open versus laparoscopic surgery in age groups.

**Figure 2 medicina-59-00295-f002:**
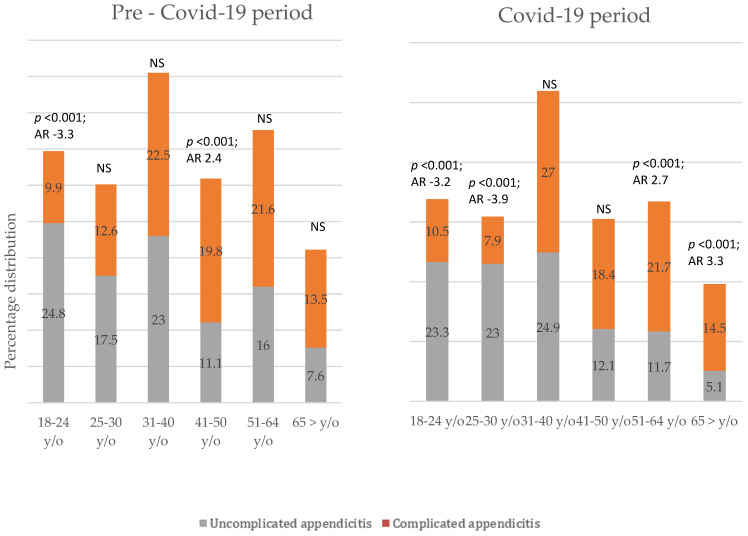
Frequency of uncomplicated and complicated appendicitis in every age group.

**Table 1 medicina-59-00295-t001:** Demographics of study population suffering acute appendicitis.

Variables	Patients during Pre-COVID-19 *n* = 454	Patients during COVID-19 *n* = 409	*p* Value
	*n* (%)	*n* (%)	
Age, median (IQR)	35 (51–26)	35 (49–27)	NS
Age groups			
18–24	96 (21.15)	76 (18.58)	NS
24–30	74 (16.30)	71 (17.36)	NS
31–40	104 (22.91)	105 (25.67)	NS
41–50	60 (13.22)	59 (14.43)	NS
50–64	79 (17.40)	63 (15.40)	NS
>65	41 (9.03)	35 (8.56)	NS
Gender			
Male	218 (48)	201 (49.1)	NS
Female	236 (52)	208 (50.9)	NS
Severity of appendicitis			
Uncomplicated appendicitis	343 (75.6)	257 (62.8)	<0.001
Complicated appendicitis	111 (24.4)	152 (37.2)	<0.001

NS: not significant.

**Table 2 medicina-59-00295-t002:** Comparison of preoperative diagnostic modalities.

	Patients during Pre-COVID-19 *n* = 454	Patients during COVID-19 *n* = 409	*p* Value
	*n* (%)	*n* (%)	
US	324 (71.4)	275 (67.2)	NS
CT	15 (3.3)	19 (4.6)	NS
US and CT	115 (25.3)	115 (28.1)	NS

US: ultrasound, CT: computed tomography, NS: not significant.

**Table 3 medicina-59-00295-t003:** Waiting time until surgery depending on radiological diagnostic modalities used.

Time	Patients during Pre-COVID-19 *n* = 454	Patients during COVID-19 *n* = 409	*p* Value
	*n* (%)	*n* (%)	
	Patients that had only US		
<6 h	162 (50)	149 (54.2)	NS
6–24 h	151 (46.6)	118 (42.9)	NS
>24 h	11 (3.4)	8 (2.9)	NS
	Patients that had only CT scan		
<6 h	7 (46.7)	10 (52.6)	NS
6–24 h	6 (40)	7 (36.8)	NS
>24 h	2 (13.3)	2 (10.5)	NS
	Patients that had US and CT scan		
<6 h	38 (33.0)	38 (33)	NS
6–24 h	60 (52.2)	63 (54.8)	NS
>24 h	17 (14.8)	14 (12.2)	NS

US: ultrasound, CT: computed tomography, NS: not significant.

**Table 4 medicina-59-00295-t004:** Hospitalization and short follow-up outcomes.

Variable	Patients during Pre-COVID-19 *n* = 454	Patients during COVID-19 *n* = 409	*p* Value
	*n* (%)	*n* (%)	
Overall hospital stay (days), median (IQR)	4 (5–3)	3 (5–3)	NS
Overall hospital stay			
<3 days	214 (47.10)	207 (50.6)	NS
4–6 days	191 (42.10)	166 (40.6)	NS
>7 days	49 (10.80)	36 (8.8)	NS
Admission to ICU (number of patients)	10 (2.2)	13 (3.2)	NS
Mortality	1 (0.2)	0 (0)	NS

NS: not significant.

**Table 5 medicina-59-00295-t005:** Correlation analysis between patients age and overall hospital stay.

	Pre-COVID-19 Period		COVID-19 Period	
	R	*p* Value	R	*p* Value
	All patients			
Overall hospital stay	0.312	<0.001	0.348	<0.001
	Uncomplicated appendicitis			
Overall hospital stay	0.191	<0.001	0.185	0.003
	Local peritonitis			
Overall hospital stay	0.405	<0.001	0.396	<0.001
	Generalized peritonitis			
Overall hospital stays	0.48	0.06	0.497	0.012

## Data Availability

Not applicable.

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
