# Peer review of "The Course and Surgical Treatment of Acute Appendicitis during the First and Second Wave of the COVID-19 Pandemic: A Retrospective Analysis in University Affiliated Hospital in Latvia"

_medicina, 2023, doi:10.3390/medicina59020295_

Round 1
Reviewer 1 Report
The paper titled " The Course and Surgical Treatment of Acute Appendicitis During the COVID-19 Pandemic: A Retrospective Analysis in University Affiliated Hospital in Latvia " provides a comparison of acute appendicitis management before and during the first year of of the pandemic (First and second waves).
As the study provided an interesting comparison and reflected an increase in the number of complicated appendicitis during the COVID period studied (2020), a more interesting analysis would have been conducted if the period included the later phases or waves of the pandemic and patients who tested positive for COVID. Was there data collected for the periods 2021. 2022?
The article is deemed suitable for publication pending some revisions and recommendations. Below are my comments and suggested points to be reconsidered:
1- Complicated vs uncomplicated appendicitis: The medical terms are clearly differentiated but not sufficiently graded. Please using a scaling system to classify the acute appendicitis such as that proposed by Gomes et al (grades such as 1-2: non-complicated; grades 3-5: complicated) to get a sense of how the cases where distributed.
2- The fact that the COVID period spans only 2020 is a bit misleading when you use the title (COVID period) since it only represents early phases of the pandemic. Can you please elaborate on why only 2020 was chosen and not 2020 and 2021?
Can you also describe briefly on how the pandemic progressed? How the waves hit Latvia and what were the dates of each wave approximately?
Please also modify the title and text to clearly reflect this point
3-The fact that all 409 patients admitted for appendicitis during the "COVID period" were COVID negative reflects a 0% infection rate within this group.
Can you please indicate the overall infection rate in Latvia during the coarse of the pandemic in 2020 in Latvia with reference to public available data?Was the testing only based on Rapid testing or PCR testing or both?
4- Lines 236-238 in the discussion: "The overall numbers of acute appendicitis patients however dropped due to the need of freeing up capacity of our hospital for Covid-19 patients and transferring a part of the emergencies to other hospitals."
Please run a statistical test to compare the number 454 vs 409
Thanks for your efforts in putting this work together.
Reviewer 2 Report
This is a retrospective analysis of the course and treatment of AA in an University Hospital during COVID-19, compared to a timely course before the pandemic period. It clearly is not a novel idea to investigate this question, there is a lot data/papers on exact this subject available. However, I find the presentation of the work is very nicely done and a lot of patients were included.
Introduction: Very detailed, very extensive. Please shorten it a bit.
Methods: Very nicely described
Results: Please consider to delete table 3: It can be described in one sentence but there are too many tables/graphs that are not really needed/helpful to be shown in tables. The same is for Table 4 -the content is being described in the two sentences before. Furthermore, is figure 2 needed? It is a bit overloaded with information, which is being repeated.
The discussion I find clear, structured.
Overall it is a retrospective study, with the limitations that it is not a new, innovative subject, therefore all results are displayed in different forms (written, tables, graphs). However, it is very well written and a lot of patients have been included.
